# Morphine Perinatal Exposure Induces Long-Lasting Negative Emotional States in Adult Offspring Rodents

**DOI:** 10.3390/pharmaceutics14010029

**Published:** 2021-12-24

**Authors:** Nair C. F. Castro, Izabelle S. Silva, Sabrina C. Cartágenes, Luanna M. P. Fernandes, Paula C. Ribera, Mayara A. Barros, Rui D. Prediger, Enéas A. Fontes-Júnior, Cristiane S. F. Maia

**Affiliations:** 1Laboratório de Farmacologia da Inflamação e do Comportamento, Faculdade de Farmácia, Universidade Federal do Pará, Belém 66075-900, Brazil; nair_correia@yahoo.com.br (N.C.F.C.); farmacamoes@yahoo.com.br (I.S.S.); sabrina_decarvalho@yahoo.com.br (S.C.C.); paularibera17@gmail.com (P.C.R.); mayaraarouckb@gmail.com (M.A.B.); efontes@ufpa.br (E.A.F.-J.); 2Departamento de Ciências Morfológicas e Fisiológicas, Centro das Ciências Biológicas e da Saúde (CCBS), Universidade Estadual do Pará, Belém 66087-662, Brazil; luannafe@hotmail.com; 3Departamento de Farmacologia, Centro de Ciências Biológicas, Universidade Federal de Santa Catarina, Florianópolis 88040-900, Brazil; rui.prediger@ufsc.br

**Keywords:** morphine, pregnancy, lactation, anxiety, depression, nociception

## Abstract

Psychoactive substances during pregnancy and lactation is a key problem in contemporary society, causing social, economic, and health disturbance. In 2010, about 30 million people used opioid analgesics for non-therapeutic purposes, and the prevalence of opioids use during pregnancy ranged from 1% to 21%, representing a public health problem. This study aimed to evaluate the long-lasting neurobehavioral and nociceptive consequences in adult offspring rats and mice exposed to morphine during intrauterine/lactation periods. Pregnant rats and mice were exposed subcutaneously to morphine (10 mg/kg/day) during 42 consecutive days (from the first day of pregnancy until the last day of lactation). Offspring were weighed on post-natal days (PND) 1, 5, 10, 15, 20, 30, and 60, and behavioral tasks (experiment 1) or nociceptive responses (experiment 2) were assessed at 75 days of age (adult life). Morphine-exposed female rats displayed increased spontaneous locomotor activity. More importantly, both males and female rats perinatally exposed to morphine displayed anxiety- and depressive-like behaviors. Morphine-exposed mice presented alterations in the nociceptive responses on the writhing test. This study showed that sex difference plays a role in pain threshold and that deleterious effects of morphine during pre/perinatal periods are nonrepairable in adulthood, which highlights the long-lasting clinical consequences related to anxiety, depression, and nociceptive disorders in adulthood followed by intrauterine and lactation morphine exposure.

## 1. Introduction

The use of psychoactive drugs affects individuals in modern society, causing a negative impact on society, the economy, and health. In 2013, 9.4 percent of Americans, aged 12 or older, suffer the effects of illicit drugs [1]. Globally, it is estimated that 153–300 million people aged 15–64 have used an illicit drug at least once in their lifetime, and about 12% of users develop dependence and become frequent drug users [2]. According to World Health Organization (WHO), around 250 million people die each year from diseases related to overdose or drug abuse [3,4,5].

The therapeutic use of morphine is recognized as an important analgesic, constituting the standard drug for the treatment of severe pain, multiple traumas, and anesthetic procedures [6]. Albeit opioids play an essential role in relieving pain, the problems related to tolerance and dependence restrict its use [7]. Of note, the overall prevalence in the use of opioids for non-therapeutic purposes in 2010 was estimated at around 36 million consumers aged 15–64, particularly heroin and morphine [4,5]. 

The use of opioids during pregnancy is prevalent, which reaches rates ranging from 1% to 21%. According to the National Survey on Drug Use and Health [1], 4.4% of pregnant women reported drug misuse in the last 30 days, of which 1% used opioids for non-therapeutic purposes. Opioids circulating in maternal blood induce withdrawal symptoms or even fetal overdose. Pregnant women exposed to psychotropic substances abuse develop negligent behavior, malnutrition, lack of adequate obstetric care, among other health problems [8,9,10]. 

Offspring exposed to morphine abuse during pregnancy increases the risk of miscarriage, prematurity, low birth weight, reduced length of the newborn, decreased head circumference, sudden infant death syndrome (SIDS), growth deficiency postnatal, anxiety, feelings of rejection, as well as delay in cognitive function [11]. Besides, the newborn can undergo neonatal abstinence syndrome (NAS) in the first days after birth. Characterized by negative symptoms related to the central nervous system (CNS), as well as respiratory and digestive systems dysfunction, it was observed in NAS sneezing, lacrimation, impaired sucking, reduced sleep duration, and irritability, which leads to physiological stress, requiring intensive care service, as well as contributing to low weight birth [8,12,13].

CNS damage may result in behavioral shortcomings rather than physical defects. The long-term neurophysiological changes are associated with impaired intellectual ability and emotional control in children, including hyperactivity [14,15,16,17,18]. Actually, prenatal exposure to morphine affects several CNS structures, which may contribute to psychiatric disorders (for review see [19]).

Concerning early-life opioid exposure toxicological effects related to nociceptive responses, the findings are controversial. For instance, Tempel et al. [20] described that prenatal exposure to morphine, at doses of 75 mg on gestation day (GD) 16, induces tolerance responses to analgesic effects in rats, through the downregulation of opioid receptors. In contrast, Tao et al. [13] postulated that intrauterine morphine administration at a dose of 2 mg/kg alters pain threshold, inducing hyperalgesia, mainly in female rats.

Studies revealed that µ-opioid receptors occur in the early embryonic stages [21]. The opioid system modulates a variety of neurotransmitters, including noradrenaline, which presents a fundamental role in the neurodevelopment processes [22]. In rodents, noradrenergic neurons emerge at GD10-13, the cortical innervation occurs at GD16-17, the relevant synaptogenesis of noradrenergic signaling occurs at PND10-15 and PND20–30, while noradrenaline adult levels solely take place at PND 35–70 (for review see [22]). In this sense, abnormal activation of opioid signaling during the early stages of pregnancy may affect the noradrenaline pathway, resulting in long-lasting behavioral disorders, also called hyperkatifeia [23,24].

Considering the increased prevalence of opioid abuse by pregnant women and their consequences still poorly elucidated, we decided to investigate the long-lasting negative emotional states (hyperkatifeia) and sex differences induced by high doses of morphine exposure during pregnancy and lactation.

## 2. Materials and Methods

### 2.1. Animals

Two-month-old Wistar rats, nulliparous females (*n* = 10) and males (*n* = 5) obtained from the Animal Facility, Biological Sciences Institute, Federal University of Pará (UFPA), and Swiss females (*n* = 10) and males (*n* = 2) mice obtained from the Instituto Evandro Chagas (IEC, Belém, Brazil) were used in the experiment and kept in collective cages (5 animals per cage). Animals were maintained in a climate-controlled room with a 12-h reverse light/dark cycle (lights ON 7:00 AM), and food and water ad libitum. All procedures were approved by the Ethics Committee on Experimental Animals of the Federal University of Pará under license number BIO-049-12 and followed the NIH Guide for the Care and Use of Laboratory Animals.

### 2.2. Experimental Design

Males and female rats and mice (couple) were placed in isolated cages protected internally by a grid to avoid contact of feces/urine and facilitate the location of vaginal plugs released by females after copulation. The presence of a vaginal plug was evaluated in the morning, indicating pregnancy. The pregnant females were isolated in cages properly identified with the possible date of birth.

According to the protocol described by Klausz et al. [25], on the first day of pregnancy (gestational day (GD)1), females were randomly grouped and treated daily with morphine (10 mg/kg/day, subcutaneously-s.c.) or saline (s.c.; control group) until the last day of lactation 9postnatal day (PND)210. This high dose was chosen considering the misuse of morphine may exceed the therapeutic recommendations (100 mg/day). In addition, allometric extrapolations indicated that the dose of 10 mg/kg administered in animals is equivalent to 2.31 mg/kg in humans (70 kg adult male), in a final dose of 161.7 mg/day). Besides, subcutaneous via presents bioequivalence to intravenous serum levels [26]. Finally, we expected that high morphine serum levels provide increased morphine milk levels, considering that morphine milk levels can reach values higher than those observed in plasma levels [27,28].

Offspring weight gain was monitored every 5 days until 30 and 60 days old. To prevent handling stress during the lactation period (D1-D20), a sample (*n* = 5) of each experimental group was weighted. Following PND21, offspring were sexed and kept in cages (39 × 32 × 16 cm), under standard conditions of temperature, light/dark cycle control, food, and water ad libitum until behavioral and nociception tests [75th postnatal day (PND75)]. To minimize the litter effect, 1 animal/sex/litter was selected for behavioral assays on PND21. Animals that were not selected were conducted to euthanasia. Researchers were blinded to all stages of the study, including data analysis.

#### 2.2.1. Experiment 1

This stage of the work occurred to evaluate the neurobehavioral sequelae resulting from the neurodevelopmental period exposed to morphine. Behavioral tests were conducted with rats at PND75 and consisted of open field (OF), elevated plus maze (EPM), and forced swim test (FST) in a sequential battery. Individuals that fell from the equipment during the behavioral assays were excluded from the respective test.

##### Open Field (OF)

Briefly, animals were individually placed for 5 min at the center of the acrylic black arena (100 × 100 × 45 cm), divided into 25 squares (20 × 20 cm). The numbers of total squares crossed, central area time (central area time/300 s × 100; %), and rearing were recorded [29]. Total squares crossed were adopted to evaluate the level of ambulation of the offspring. The central area time was chosen to measure the anxiety-like behavior in open spaces, as the arena. The rearing was adopted to assess the vertical exploration.

##### Elevated Plus-Maze (EPM)

Following the OF performance, animals were tested in EPM test that consists of a plus-shaped wooden maze with two opposite open arms (50 × 10 cm) and two enclosed arms (50 × 10 × 40 cm), spreading out from a central platform (10 × 10 cm), elevated 50 cm from the floor. According to the protocol described by Pellow et al. [29], animals were individually placed in the center of EPM, facing one of the enclosed arms. Free exploration of the apparatus was allowed for 5 min. The number of enclosed arms entries (EAE) and percentage of open arms time (%OAT), which was calculated according to the formula: open arms time/total time × 100, were evaluated. Anxiogenic effects are defined as a decrease in time spent on open arms. The number of enclosed arms entries (EAE) was used as the parameter of locomotor activity in the task [30].

##### Forced Swimming Test (FST)

Following the EPM performance, animals were submitted to FST. Briefly, rats were placed in a Plexiglas cylinder (diameter 30 cm, height 50 cm), filled with water (high 40 cm) at a temperature of 23 ± 1 °C for 5 min. The immobility time was recorded during the last 3 min. It was considered immobility when the animal remained floating on the water with minimal movements necessary to keep its head above the water [31]. Increased immobility time was admitted as depressive-like behavior [31].

#### 2.2.2. Experiment 2

To prove the hypothesis that perinatal exposure may display long-lasting alterations on the nociception profile, we conducted the writhing test nociception assay in mice.

##### Acetic Acid-Induced Writhing Test

According to the protocol described by Koster et al. [32], this method evaluates antinociceptive activity through nociception induction by chemical stimulation (0.1 mL/10 g of acetic acid 0.6%, intraperitoneally). This chemical-induced nociception model presents nonselective nociception sensibility, stimulating nociceptive pathways sensible to a variety of drugs that acts centrally or peripherally, as narcotics and nonsteroidal anti-inflammatory substances, and can detect hyperalgesia states by multiple mediators, as well as exhibits characteristics similar to clinical pain in its nature [32,33]. Following the nociceptive stimulus, mice were individually placed in a glass cylinder of 22 cm diameter. The number of abdominal contractions (writhes) following 10 min of stimulus over a period of 20 min were recorded. The nociceptive response was determined as the level in the number of writhes. To minimize distress and suffering, immediately after the writhing test, animals were anesthetized and conducted to euthanasia. To avoid the variability in mice nociception induced by a male experimenter, a female researcher applied the writhing test [34].

All experimental procedures are summarized in Figure 1.

##### Statistical Analysis

Values are expressed as the mean ± standard deviation of the mean (SDM) of 8–10 animals per group (rats; male control = 10; female control = 10; male morphine = 8; female morphine = 10) and 6–10 animals per group (mice; male control = 6; female control = 6; male morphine = 8; female morphine = 10). Data distribution was assessed using the Kolmogorov–Smirnov test. Data with normal distribution were submitted to Student’s *t*-test or Two-way Analysis of Variance (ANOVA) with Tukey’s post-hoc test (as variables treatment vs. sex). Nonparametric data were evaluated using the Mann–Whitney Rank Sum test. For offspring body weight analysis, Student’s t-test was used, considering each time-point. The probability adopted as indicative of the existence of significant differences was *p* < 0.05. The graphical construction and statistical analysis were performed using Sigma Plot 12.5 software (San Jose, CA, USA).

## 3. Results

### 3.1. High Doses of Morphine Exposure during Pregnancy and Lactation Reduces Weight Gain during Childhood and Adolescence, but Not Adulthood

Weight gain of rats is summarized in Figure 2. Animals exposed to high doses of morphine during pregnancy and lactation reduced body weight gain during adolescence, at D5, D10, D15, D20, D30 days of life (D5: *p* = 0.00423; 95% IC 1.332–6.616; D10: *p* = 0.00180; 95% IC 1.717–6.219; D15: *p* = 0.00264; 95% IC 3.050–8.350; D20: *p* = 0.0102; 95% IC 1.273–11,487; D30: *p* = 0.000925; 95% IC 11.270–34.330), that was recovery at 60 days of life (*p* = 0.944; Figure 2).

### 3.2. High Doses of Morphine Exposure during Pregnancy and Lactation Induces Hyperlocomotion, Anxiety- and Depressive-like Behavior in Adult Rats

As illustrated in Figure 3A, female rats exposed perinatally to morphine displayed locomotor activity increase in OF test (F = 9.405; *p* = 0.001), whereas this parameter was not altered in male rats (Figure 3A). In addition, such increment in spontaneous ambulation was even higher than the counterparts (F = 11.486; *p* = 0.002).

Figure 3B shows the results of the percentage of locomotion in the central “aversive” area of the OF, which represents a parameter related to anxiety-like behavior in rats. Male and female offspring exposed to high doses of morphine during pregnancy and lactation reduced the time in the central area (control vs. morphine: male, *p* = 0.00108; 95% IC 2.123–6.874; female *p* = 0.002, U = 7.000), indicative of anxiogenic-like profile. Female control individuals also reduced central area time exploitation compared to the male control group (Male vs. Females: *p* = 0.016, U = 15.000). As illustrated in Figure 3C, the number of rearing was not affected by morphine intrauterine exposure.

In EPM test, rats exposed perinatally to high doses of morphine displayed reduced %OAT (control vs. morphine: male *p* = 0.0125, 95% IC 1.013–12.499; female *p* = 0.000206, 95% IC 3.628–10.123; Figure 4A), confirming the anxiogenic-like profile observed in OF test.

The number of EAE, used as a motor control parameter in the EPM assay, was not altered in animals exposed to morphine (Figure 4B).

In FST, both male and female rats exposed to high doses of morphine during pregnancy and lactation augmented the immobility time parameter (Control vs. Morphine: male *p* = 0.00551, 95% IC 34.192–3.066; female *p* = 0.0138, 95% IC 36.028–2.631), indicative of depressive-like behavior (Figure 5).

### 3.3. High Doses of Morphine Exposure during Pregnancy and Lactation Alters Nociceptive Responses Profile in Adult Mice

In the acid acetic test, animals (regardless of sex) exposed to high doses of morphine during the perinatal period were more susceptible to a nociceptive stimulus (Control vs. Morphine: F = 31.696; male *p* = 0.001; female *p* < 0.001; Figure 6). However, female-treated mice showed higher sensibility to chemical noxious stimuli than males (male vs. female: F = 8.085; *p* = 0.022).

## 4. Discussion

In the present study, we investigated neurobehavioral and nociceptive changes in offspring exposed to high doses of morphine during intrauterine and immediate postnatal (lactation) periods. Our findings demonstrated that perinatal morphine exposure elicited long-lasting behavioral consequences, as anxiogenic- and depressive-like behaviors, as well as modification in the profile of pain sex-independent in rodents. In fact, this is the first time that high doses of morphine perinatal exposure are investigated under the entire perinatal period (from GD1 to PND21). Moreover, it is also unprecedented the investigation of long-lasting deleterious effects of high doses of morphine perinatal exposure, when animals reached the complete maturation of the CNS, on adult life (PND75), as well as the pain threshold differences between males and females. 

Firstly, there are different morphophysiological features between human and rodent placenta, which may be considered a toxicological challenge during pregnancy [35]. The relevant contrast that may interfere with teratogenic agent exchange across the placenta is related to the fetal-maternal interface, which consists of a hemochorial type with three layers in rats and one layer in humans (for review see [35]). In addition, the yolk sac differences between humans and rats limit the toxicity effects of teratogenic agents that are based on its toxicological mechanism by yolk sac dysfunction [35]. Following these relevant issues, we might consider that the lipophilicity of morphine supports its possibility of crossing biological membranes, such as the placenta, inducing the teratogenic effects through other mechanisms than yolk sac dysfunction [26].

We found that high doses of morphine administered during pregnancy and lactation induced reduction in the animals’ body weight gain from 5 to 30 days of life, which was the recovery in adulthood (60 days old). In fact, lower weight birth after morphine prenatal exposure has been reported previously at doses of 5 or 10 mg/kg [25,36], however, this issue is still contradictory at equal doses administered [37]. We hypothesize that animals in early life may suffer the consequences of reduced hepatic activity and morphine withdrawal (NAS), which reduces the colostrum and breast milk sucking and consequently body weight [11]. In addition, the effects of perinatal exposure to morphine become less pronounced only after a longer period, which may explain the absence of significant differences related to body weight at two months of age [38,39]. 

Nowadays, it is well accepted that prenatal exposure to morphine displays anxiogenic behavior in rodents (dose of 5 mg/kg followed 10 mg/kg) [39]. According to Velisek et al. [40], morphine prenatal exposure in rats induces stress conditions, both in males and females. Indeed, chronic treatment with morphine induces a state of chronic stress in mothers, causing hyperactivity of the hypothalamic-pituitary-adrenal (HPA), which in turn can reprogram the development of the HPA axis in offspring, with over secretion of glucocorticoids, characterized as important factors in psychiatric-like disorders, such as anxiety and depression [25,41,42,43,44,45]. Our results show that both female and male offspring exposed to high doses of morphine during the perinatal period display anxiety- and depression-like behavior sex-independently in adulthood, accessed by OF, EPM, and FST tests. 

Several studies have investigated neurobehavioral alterations elicited by morphine prenatal exposure. Ahmadalipour et al. [39] reported that morphine prenatal exposure from GD11 to 18 elicited anxiety-like behavior in juvenile rats. Contradictorily, Tan et al. [46] study found the anxiolytic effect in adult rats following morphine intrauterine exposure from GD9 to 18 at the same dose regimen of Ahmadalipour study [39] (first three morphine administration was 5 mg/kg, followed by subsequent injections of 10 mg/kg). Indeed, other works have demonstrated that such doses of morphine prenatal exposure may induce depressive-like effects in rodents [25,40] due to an increase of catecholamines (noradrenaline) in the hypothalamus [41] and reduced corticosteroids blood levels in adulthood [25]. In contrast, Ahmadalipour et al. [39] failed in demonstrating depressive-like behavior in the pups exposed to morphine during the intrauterine period. In other words, contradictory data have been reported about this issue due to different period and exposure times, however, few studies have explored the entire perinatal period, which includes the immediate postnatal day. Considering that the noradrenergic pathway, which is also regulated by the opioid system, consists of a fundamental agent on the neurodevelopment that emerges at the early-pregnancy period (GD10-13) and reaches the mature signaling at PND70, in which different period and times of the intervention may display specific alterations [22]. Our morphine perinatal challenge was rarely employed, except by Klausz et al. (2011) [25] that did not assess sex differences, as well as explored the behavioral alterations at PND 60 (early adulthood) in which the adult levels of NA was not still reached (PND 70) [22]. Besides, pregnancy unplanned occurs frequently among opioid-dependent women, whereupon opioid maintenance and relapse risk present high prevalence [11]. Thus, we suggest that our protocol that ranges from GD1 to PND21 may display an increase of behavioral disturbance with long-term reflects in adulthood (PND75). These neuroendocrine pathophysiology theories require further studies and comprise the limitation of the present work.

Spontaneous locomotor activity of rats perinatally exposed to high doses of morphine was also modified in our study. Exploratory behavior was increased among female, but not male, suggesting that female consists of a vulnerable group for modifications in motivational ambulation activity followed intrauterine exposure to morphine in rats. Tan et al.’s [46] study found a similar result with males, in which morphine prenatal exposure (5 mg/kg followed 10 mg/kg; GD9-18) did not affect the spontaneous locomotor profile. Unfortunately, in the Tan [46] study, females were not evaluated. Besides, the observation that the greatest concentration of neuronal opioid receptors relies on the limbic system, thalamus, striatum, hypothalamus, midbrain, and spinal cord [47] suggests that physiological mechanisms other than analgesia and pain perception may be affected by narcotics.

As opioid drugs affect the opioid pathway during neurodevelopment, we also decided to investigate the pain threshold in both genders. In the nociception assay, high doses of morphine prenatal exposure also affected the profile of pain perception. Actually, evidence suggests that males and females exhibit differences in pain susceptibility, which females seem to be more sensible in the perception of pain, as well as the present increases in negative response against a nociceptive stimulus than males [48].

Intraperitoneal injection of acetic acid represents a mixed model, which activates visceral and somatic nociceptors, elicits inflammation in the sub-cutaneous muscle of the abdominal wall, as well as in sub-diaphragmatic visceral structures [49]. Besides, this nociception paradigm represents clinical pain in comparison with its nature [33]. The substance P (SP) and its neurokinin-1 receptor (NK1) play a fundamental role in visceral and inflammatory pain, activating afferent plexus originating in gastrointestinal regions [50,51]. Morphine reduces SP release in the spinal cord through the activation of μ-opioid receptors, which in turn reduces the number of c-fos-positive neurons in superficial laminae and the pain [52]. The present study demonstrated through this worldwide accepted nociception experiment that perinatal exposure to high doses of morphine induces an increase in the susceptibility to pain in male and female individuals late in life. However, female offspring were revealed to be more vulnerable to chemical nociception stimulus, increasing the number of writhes compared to male offspring. In this sense, our results suggest that our morphine prenatal and lactation exposure model induces long-lasting alterations in the nociceptive pathway. 

Biglarnia et al. [53] postulated that short-time opioid high-dose therapy during pregnancy induces long-term sex-related alterations in pain response. These sex-dependent anti-nociceptive effects could be observed through modified receptive responses to environmental stimuli caused by alterations in pain stress-sensitive brain circuitry and opioid-receptor pathway. The authors claim that such differences are related to the rate and timing of brain development between males and females [54]. Moreover, sexual differences on neurotransmitters pathway, like dopamine and norepinephrine, during neurodevelopment presumably may also affect pain responses [55,56].

In fact, modification of profile pain during the perinatal period seems to occur through μ-opioid receptors, since this receptor is expressed during early postnatal maturation, mainly in great diameter neurons, which are present in high concentration on dorsal root [57]. During the neonatal period occurs deep structural and functional reorganization in the spinal sensory systems [57,58], which is strongly influenced by local abnormal or excessive activity in the maturation of type C nociceptive fibers [59,60]. Thus, interference during the neonatal period can lead to changes in the activity of primary afferent processing of the spinal cord and descending pain control, as well as the maturation of nociceptive circuitry at different levels of neuroaxis [59]. The increased expression of μ receptors suggests a broader action of morphine activity directly into the spinal cord and, indirectly, through the wide ends of primary afferents in young rats (PND7) [59,60,61]. We hypothesize that morphine exposure through breastfeeding at the PND7, a critical period for C-fiber nociceptive maturation, may alter the nociception pathway, affecting susceptibility to pain. However, such pain threshold modification was different between males and females.

Actually, the distinct physiological changes between males and females in the morphine intrauterine exposure model are still poorly understood. Previous studies have shown that the emergence of opioid receptors in rats is parallel to the onset of steroid receptors in the brain, which is around 14 days of fetal life when the final cell division of the hypothalamus is completed [62,63]. This period also coincides with the onset of gonadal steroid secretion in the male [64] and such events may serve as signals for the beginning of the ‘critical time’ of sexual differentiation at the central level [65]. Studies reported that the spinal cord opioid system is, in many respects, more robust and efficient in males than in female rodents. In fact, females present a lower pain threshold, as well as reduced tolerance to noxious stimuli when compared to males (see [66] for review). The midbrain periaqueductal gray (PAG) and its descending projections are the keys to the nociceptive threshold. PAG presents elevated μ-opioid receptor density, which provides analgesic effects followed by stimulation. Males seem to activate PAG and its descending projection more efficiently than females (see [66] for review), which supports our findings. In this sense, we suggest that high doses of morphine perinatal exposure may alter PAG pathway modulation, increasing pain susceptibility both in males and females in adulthood, however, more efficiently in females.

All statistical tests applied presented above of 0.727 in the power of the test. We highlight the limitations of the present work based on animal models to extrapolate to humans. Albeit there is a homology on the sequence of amino acids residues up to 90% between human and rodent opioid receptors, however peculiar differences related to acid amino residues sequence modify relevant signaling in the opioid pathway [67]. Therefore, it has been assumed that such dissimilarity permits the extrapolation of rodent-to-human translational research data in both analgesic and tolerance domains [67,68].

## 5. Conclusions

These results demonstrated that high doses of morphine exposure during pregnancy and lactation induce long-lasting negative emotional states, also called hyperkatifeia, in adult offspring rodents. Besides, morphine perinatal exposure affected nociception modulation, increasing the susceptibility to pain, mainly in females, showing that pain threshold difference is sex-specific. Besides, the hyperkatifeia induced by high doses of morphine perinatal exposure are not recovery in adulthood. However, additional studies are needed to evaluate the signaling pathways that underlie our findings.

## Figures and Tables

**Figure 1 pharmaceutics-14-00029-f001:**
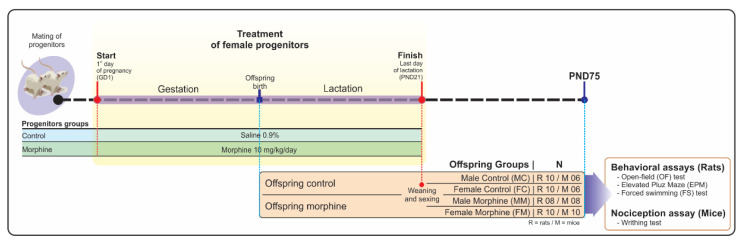
Experimental design.

**Figure 2 pharmaceutics-14-00029-f002:**
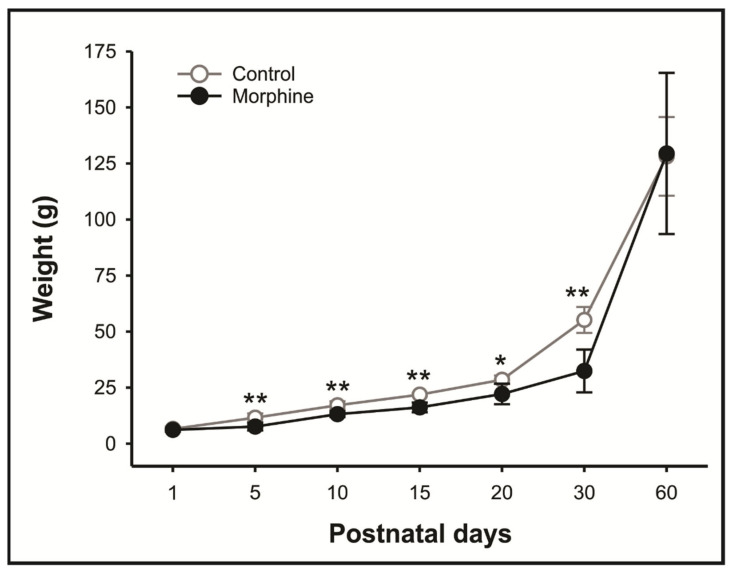
Effect of morphine treatment (10 mg/kg/day) during pregnancy and lactation on rat offspring body weight. Results expressed as mean ± S.D.M. (*n* = 5). * *p* < 0.05 morphine vs. control group; ** *p* < 0.01 morphine vs. control group. Student-t test.

**Figure 3 pharmaceutics-14-00029-f003:**
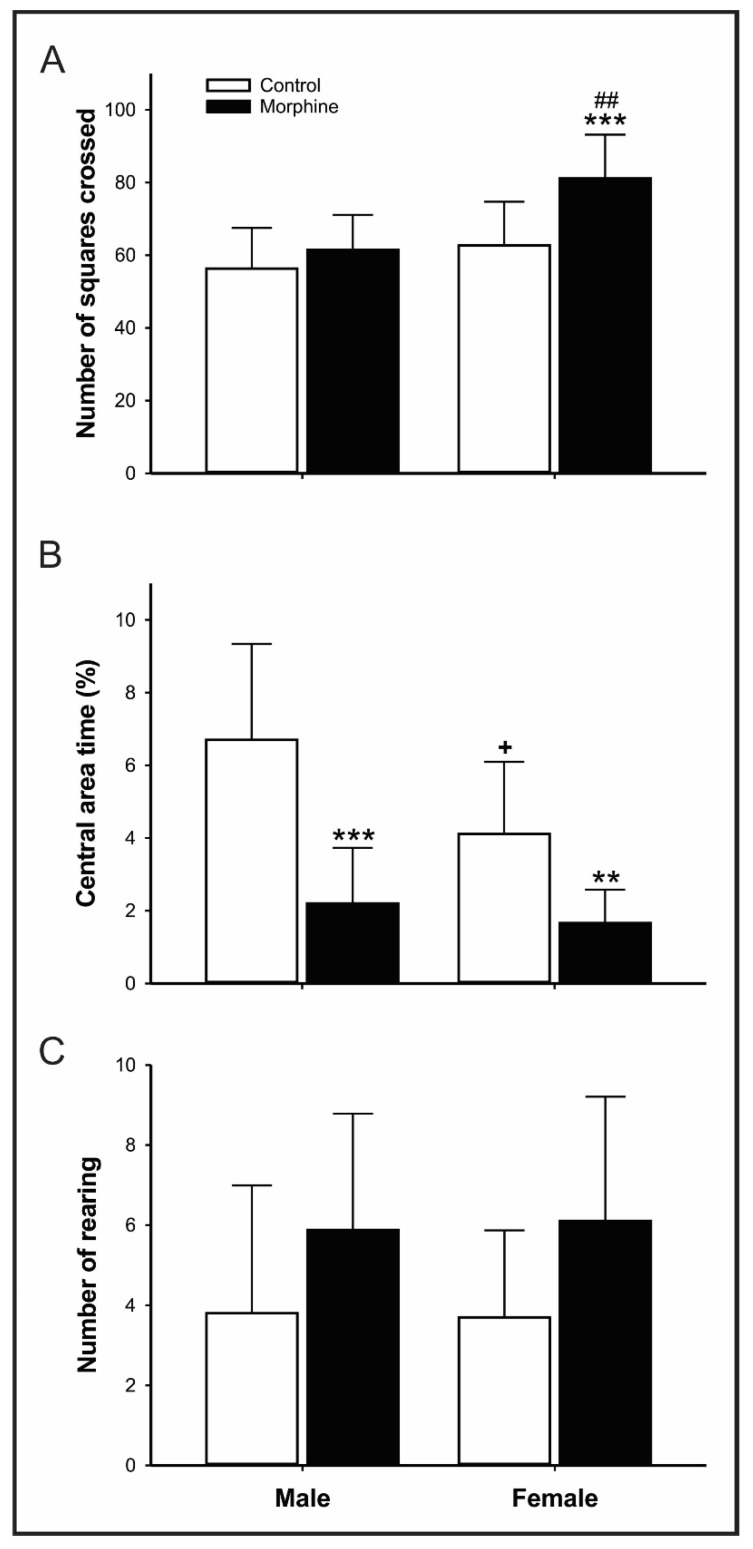
Effect of morphine treatment (10 mg/kg/day) during pregnancy and lactation on rat offspring (**A**) spontaneous locomotion; (**B**) time of central area locomotion; (**C**) number of rearing in the open field test for 5 min. Results expressed as mean ± S.D.M. (male control = 10; female control = 10; male morphine = 8; female morphine = 10). ** *p* < 0.01 morphine vs. respective control group. *** *p* < 0.001 morphine vs. respective control group. ^+^
*p* < 0.05 female control vs. male control group. ^##^
*p* < 0.01 female morphine vs. male morphine group. Two-way ANOVA followed by Turkey’s post hoc test (**A**,**C**) or Student’s *t*-test/Mann–Whitney test (**B**).

**Figure 4 pharmaceutics-14-00029-f004:**
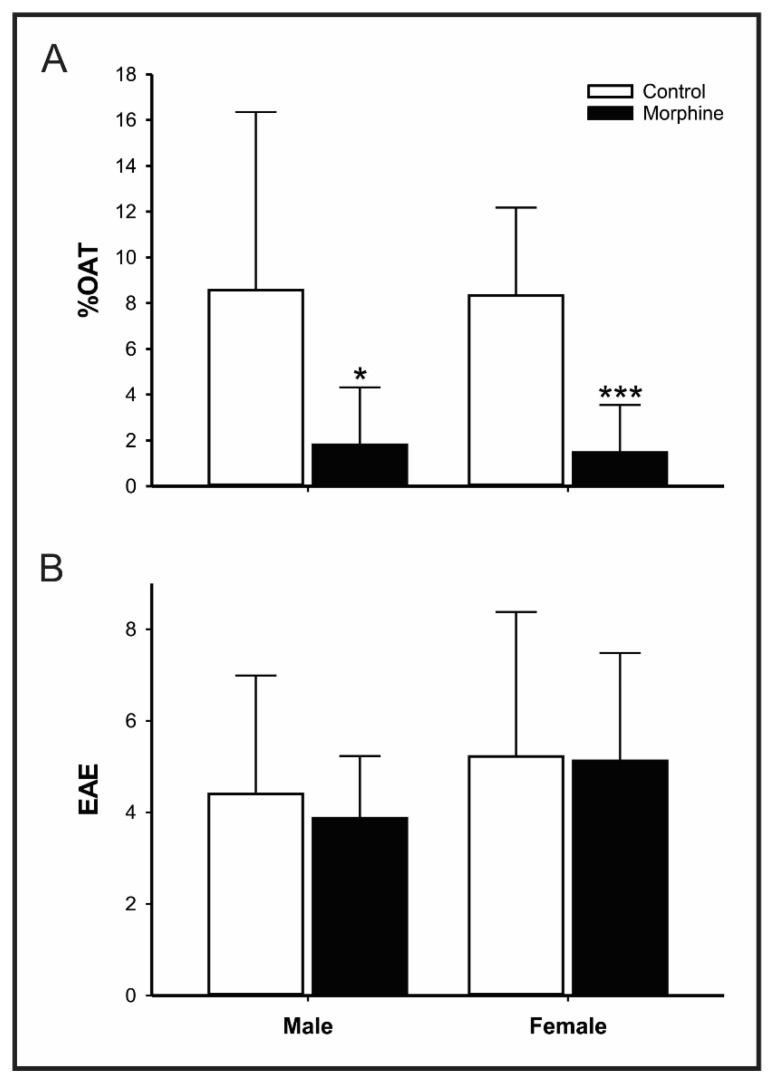
Effect of morphine treatment (10 mg/kg/day) during pregnancy and lactation on rat offspring (**A**) percentage of time in open arms; and (**B**) number of entries in the enclosed arms (EAE) in Elevated Plus Maze test. Results expressed as mean ± S.D.M. (male control = 10; female control = 10; male morphine = 8; female morphine = 10). * *p* < 0.05 morphine vs. respective control group. *** *p* < 0.001 morphine vs. respective control group. Student’s *t*-test.

**Figure 5 pharmaceutics-14-00029-f005:**
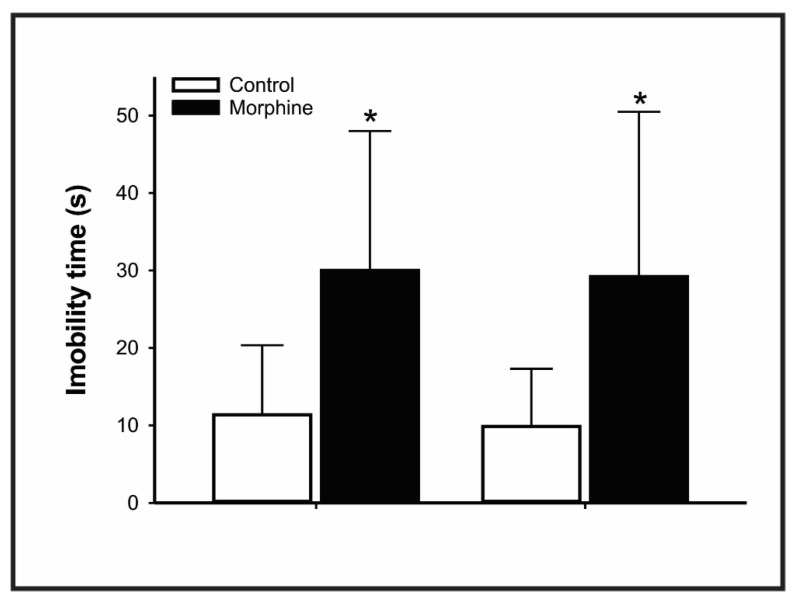
Effect of morphine treatment (10 mg/kg/day) during pregnancy and lactation on rat offspring immobility time in the forced swimming test. Results expressed as mean ± S.D.M. (male control = 10; female control = 10; male morphine = 8; female morphine = 10). * *p* < 0.05 morphine vs. respective control group. Student’s *t*-test.

**Figure 6 pharmaceutics-14-00029-f006:**
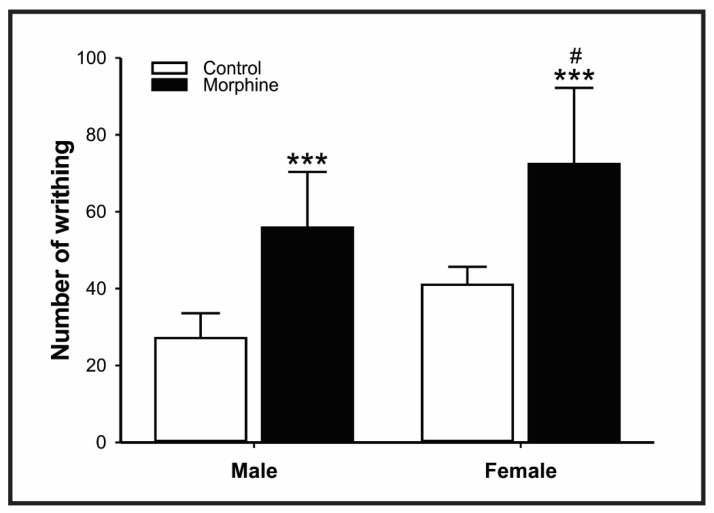
Effect of morphine treatment (10 mg/kg/day) during pregnancy and lactation on mice offspring number of writhing acetic acid-induced. Results expressed as mean ± S.D.M. (male control = 6; female control = 6; male morphine = 8; female morphine = 10). *** *p* < 0.001 morphine vs. respective control group. ^#^
*p* < 0.05 female morphine group vs. male morphine group. Two-way ANOVA followed by Turkey’s post hoc test.

## Data Availability

All data available are reported in the article.

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
