# Peer review of "Morphine Perinatal Exposure Induces Long-Lasting Negative Emotional States in Adult Offspring Rodents"

_pharmaceutics, 2021, doi:10.3390/pharmaceutics14010029_

Round 1

Reviewer 1 Report

The authors established a morphine intrauterine and lactation animal model and investigated the neurobehavioral and nociceptive changes on the offspring until PND 75. In general, this study and results are important for the opioid abuse field. However, the experiment designs are slightly inadequate. The authors need to provide more advantages and novel results to enhance the clarity and integrality of their behavioral findings and hypotheses.

Concerns are listed below:

  1. Morphine prenatal exposure could induce anxiety, depression, and gender differences in offspring have been reported by many clinical and basic studies. Although the authors indicated that their treatment schedule is the entire perinatal period (from GD1 to PND21). According to the results, I think this schedule to compare with the schedule of intrauterine exposure only may not be different on offspring, and it could also not mimic the human cases. In the situation of human, the fetus from heroin-abuse mother usually have neonatal abstinence syndrome (NAS), and the morphine is a therapeutic medicine for NAS fetus, but the doses will continue decreases.
  2. How to avoid the litter effect? It could not be prevented according to the current experimental designs.
  3. Although authors provided that many references of neuroendocrine molecules and neurotransmitters changed to explain their behavioral findings, and emphasized their results are contradictory with some previous studies. The critical question arises: Are the neuroendocrine molecules and neurotransmitters of the current treated schedule are really alterative and different from the past contradictory findings? More biochemistry or cellular experiment should be conducted to confirm the authors’ hypotheses.
  4. Why did the authors change the animal model from rat to mouse in the acetic acid-induced writhing test?

Reviewer 2 Report

The aim of the study was to assess the effects of perinatal morphine exposure on behavior and nociceptive responses in adult offsprings. The working hypothesis is actual, as opioid abuse in pregnancy is a current health system problem.

The major drawback of the study is the dose tested - 10 mg/kg/day to rats – which substantially exceeds the dose range exposure in human population, and therefore the results translation is quite vague. The experimental design should include dose range, which is more adequate for the results translation to human population (also small, moderate and high dose of morphine should be used to address/define morphine effects). In human situation, postnatal infants are exposed via milk, and due to high first pass effect of morphine the infant/newborn opioid exposure is substantially lower than proposed by authors (if addicted mothers breastfed a newborn).

Why the authors used the writhing test nociception assay for opioid agents characteristics. The test is rather recommended for testing of peripherally acting drugs. The flick tail or hot plate tests approach are of preference to study opioid analgesics. Those test should be added to the protocol to support the discussed mechanisms.

The experimental design include one time point assay, which reflects a sum of intrauterine and postnatal opioid exposure. Why the authors did not analyze both aspects (and may be as the third option a sum of them – pre- and postnatal exposure).

Minor:

The values should be presented as mean ± standard deviation (not error).

In the discussion section general findings are reported, but doses/morphine exposure were variable in different specified published papers.

English should be smoother, some imperfections are present.

Reviewer 3 Report

The manuscript by Castro and colleagues describes original findings about the role of perinatal morphine, which are leading to long-lasting effects. The paper is interesting and I have listed my comments below.

It could be interesting to evaluate the behavioral alterations induced by morphine exposure over time. Why did the Authors assess the effects of opioids exposure only once instead of at different time points?

Did the Authors perform an a priori calculation of sample size for their experiments? For example, figure 3 shows that morphine exposure resulted in significant changes of spontaneous locomotion and time of central area locomotion, but not in the number of rearing in the open field test, despite a non-significant difference seems to be present.

For ethical reasons, it should be clearly stated what happened to animals after the writhing test. 

Reviewer 4 Report

Line 34-45 please avoid numerous information about statistics, make more introduction about morphine distribution by milk and placenta. Please remember that all findings related to placental transfer in rats model should be done very cautiously https://www.ncbi.nlm.nih.gov/pmc/articles/PMC6361663/

„Considering the increased prevalence of opioids abuse by pregnant women and their consequences still poorly elucidated, we hypothesize that morphine exposure during neu- rodevelopmental period induces long-lasting impairment eliciting 1) neuropsychiatric  disorders; and 2) nociception alterations in the adulthood.”

Please trye rewording the sentence and present clear aim of the study rather than pure hypothesis.

“treated daily with morphine [10 mg/kg/day, subcutaneously (s.c.)] ”

Daily doses morphine sulphate will not usually exceed 100 mg per day in adults and adolescents over 12 years. 100 mg per 70 kg BW daily = 1.43 mg/kg (humans; https://www.medicines.org.uk/emc/product/6426/smpc#gref)

Its not clear why 10 mg/kg/day was used in rat model. How the dose was recalculated between models human versus rat. FDA state (https://www.fda.gov/media/72309/download) that if we want recalculate rat dose into human equivalent dose we should multiply rat dose by 0.16 it means that current rat model gives 10 x 0.16 = 1.6 mg/kg/ per day (rat) versus 1.43 mg/kg (humans). The calculation is valid only in case if SC bioavailability of morphine in rats is = 100%.  Please explain in the paper in context of SC bioavailability to make readers sure how translate is the model.

Please show in all figures standard deviation not SEM.

“Firstly, we found that morphine exposed during pregnancy and lactation promoted 248 reduction in the animals’ body weight gain from 5 to 30 days of life, that was recovery in 249 the adulthood (60 days old). In fact, lower weight birth after morphine prenatal exposure 250 has been reported previously [26,21], however, this issue is still contradictory”

Please explain in context mentioned in introduction and related to impaired sucking loss of colostrum etc..

Please explain model weakness in discussion especially based on differences related to opioid receptors expression and affinity differences, rats versus humans.

“Results expressed as mean ± S.E.M. (n = 8-10).”

Its unclear 8-10 if it means the observation was done for 8,9 and 10 please correct for n=3. Please make necessary corrections across manuscript.

Please add information about number of animals taken for final statistical analysis in Figure 1 for all groups for example “Male Morphine (MM) (n=xxxx)” “Control (n=xxxx)”

Please explain in discussion section power of the study and weakness related to pilot nature and  bases for sample size of the study.

“Values are expressed as the mean ± standard error of the mean (SEM) of 8-10 animals”

please show exact number of animals not range, in current form its not clear.

Reviewer 5 Report

This paper is a first in field neurobehavioral and nociceptive study of rat offspring exposed to morphine during intrauterine and postnatal periods, up to adulthood. The experiments were conducted robustly, but I fear that the group has reached conclusions already present in the literature (as cited by the paper). The two novel findings are that sex difference plays a role in pain threshold, and that deleterious effects of morphine during pre/perinatal periods are nonrepairable in adulthood.

Comment on abstract: the novelty of the study comes from its duration. Nowhere in the abstract, or the introduction is this mentioned.  

Experiment 1, lines 116-120 – EPM is used to measure anxiogenic potential of morphine as a locomotor activity modulator. Analogously, the reason why number of rearing were recorded is not specified at this point (mentioned later in line 198).

(Line 252 – 253) isn’t reduced hepatic activity post-delivery physiologically normal?

(Line 267 – 268) I do not understand where the contradiction lies between the two cited studies. Both investigators reached the same conclusion looking at different time periods. Maybe authors are referring to findings in Line 273.

(Line 276 – 277) I do not understand what the authors mean by “we believe our protocol may display an increase of behavioral disturbance with long-term reflects in adulthood.”

(Line 283 – 284) Authors again call a finding by Tan et al. “contradictory”. What is written does not point to contradictory data/findings.

(Line 302) Authors do not mention the variability that is often seen with the writhing test. In addition, the presence of male experimenters is a confounding variable. (Sorge et al., 2014)

The key conclusion of the paper seemingly lies in the last paragraph of the discussion. Doesn’t it merit a more robust discussion?  

Figures 3-6 all have a two group comparison, yet a two-way ANOVA was used to calculate statistical differences, rather than (what seemingly is) the correct approach: a non-parametric unpaired test: the Wilcoxon rank sum test (aka the Mann-Whitney U-test).

Round 2

Reviewer 1 Report

The authors have addressed all my concerns. The authors have provided more direct references, evidence, and discussions to improve their hypotheses and complete the story. I acknowledge the efforts put by the authors to integrate all the suggestions made by the reviewers. The manuscript is now in an acceptable format.

Reviewer 2 Report

The authors addressed to the comments and suggestions specified in the review. However, the implemented corrections and amendments to the manuscript do not substantially improved quality of the submission.